# Nasopharyngeal Carriage of *Streptococcus pneumoniae* in Tunisian Healthy under-Five Children during a Three-Year Survey Period (2020 to 2022)

**DOI:** 10.3390/vaccines12040393

**Published:** 2024-04-09

**Authors:** Nourelhouda Ben Ayed, Sonia Ktari, Jihen Jdidi, Omar Gargouri, Fahmi Smaoui, Haifa Hachicha, Boutheina Ksibi, Sonda Mezghani, Basma Mnif, Faouzia Mahjoubi, Adnene Hammami

**Affiliations:** 1Laboratory of Microbiology, Research Laboratory for Microorganisms and Human Disease LR03SP03, Habib Bourguiba University Hospital, University of Sfax, Sfax 3029, Tunisia; sonia_ktari2002@yahoo.fr (S.K.); omar.gargouri@medecinesfax.org (O.G.); smaouifahmi@yahoo.fr (F.S.); haifa.hachicha1992@gmail.com (H.H.); ksibi-taina@outlook.com (B.K.); maalejsenda69@gmail.com (S.M.); basmamnif@gmail.com (B.M.); mahjoubi_faouzia@medecinesfax.org (F.M.); hammami.adnene@gmail.com (A.H.); 2Faculty of Medicine of Sfax, University of Sfax, Sfax 3029, Tunisia; jdiditrabelsijihen@gmail.com; 3Community Health and Epidemiology Department, Hedi Chaker University Hospital, University of Sfax, Sfax 3029, Tunisia

**Keywords:** *Streptococcus pneumoniae*, pneumococcal carriage, healthy children, serotype, antimicrobial resistance

## Abstract

We aimed to assess the prevalence of nasopharyngeal pneumococcal carriage and to determine serotype distribution, antibiotic susceptibility patterns, and evolutionary dynamics of *Streptococcus pneumoniae* isolates in healthy under-five children. Nasopharyngeal swabs were collected from healthy children over three survey periods between 2020 and 2022. All pneumococcal isolates were serotyped and tested for antimicrobial susceptibility. A total of 309 *S. pneumoniae* isolates were collected, with an overall prevalence of nasopharyngeal pneumococcal carriage of 24.4% (CI95%: [22–26.8%]). These isolates were classified into 25 different serotypes. The most common serotypes were 14 (14.9%), 19F (12%), 6B (10.4%), and 23F (7.4%), which are covered by the PCV10 vaccine, as well as 19A (8.4%) and 6A (7.8%), which are covered by the PCV13 vaccine. A significant decrease in the proportion of serotype 19F (*p* = 0.001) and an increase in serotypes 19A (*p* = 0.034) and 6A (*p* = 0.029) were observed between the three survey periods. Multidrug resistance (MDR) was noted for 56.6% of the isolates. A significant association with antimicrobial resistance was observed for the most frequent serotypes, mainly serotype 19A. In conclusion, one-quarter of healthy under-five children in Tunisia carried *S. pneumoniae* in their nasopharynx. A dominance of vaccine serotypes significantly associated with antimicrobial resistance was recorded.

## 1. Introduction

*Streptococcus pneumoniae* is an important bacterial pathogen that causes various clinical infections, including invasive pneumococcal diseases (IPDs) such as bacteremia and meningitis. Despite the overall success of vaccine programs, *S. pneumoniae* remains a major worldwide cause of morbidity and mortality, particularly among those at the extremes of age [1].

*S. pneumoniae* nasopharyngeal (NP) colonization is a prerequisite for pneumococcal disease [2]. The colonizing pneumococci can spread from the nasopharynx to surrounding tissue, causing non-IPDs, such as acute otitis media and sinusitis. Additionally, they can invade the bloodstream, which can lead to severe and potentially fatal IPD [3]. The carriage of pneumococcus precedes disease and is the reservoir and source of the horizontal spread of this pathogen between individuals [2,3]. Hence, pneumococcal NP isolates reflect the strains circulating in the community and may provide information about those potentially pathogenic [3].

The virulence of *S. pneumoniae* is largely due to its polysaccharide capsule. The substantial level of capsule diversity makes pneumococcus a highly successful pathogen. Based on the composition of the capsular polysaccharide, over 100 pneumococcal serotypes have been identified [4].

The vaccination of children has proven to be a highly effective strategy in preventing pneumococcal infections, providing direct protection to vaccinated youth, and offering indirect protection to unvaccinated individuals [5]. Since 2000, the 7-, 10-, and 13-valent pneumococcal conjugate vaccines (PCVs) have been included in routine childhood vaccination programs worldwide [5]. PCV can only protect against a limited number of pneumococcal serotypes included in the vaccine, known as vaccine serotypes (VTs) [5,6]. Following the widespread use of PCVs, an increasing trend in the frequency of NP carriage and pneumococcal infections due to pneumococci belonging to non-VTs has been observed [5,6]. This serotype replacement, which affects the magnitude of vaccine effectiveness, is a major challenge in making and using PCVs. To expand serotype coverage, two new vaccines, 15-valent PCV and 20-valent PCV, have been licensed for use. PCV10 was introduced into the national immunization program in Tunisia in April 2019. Pneumococcal vaccination is administered in a 2 + 1 schedule, with two primary doses given at two and four months and a booster dose at 11 months without additional catch-up immunization. Before this introduction, PCV7, PCV10, and PCV13 were only available in the private sector as voluntary vaccination, and the immunization rate was low.

*S. pneumoniae* is naturally susceptible to most antibiotics active against Gram-positive bacteria. However, acquired pneumococcal antibiotic resistance is increasing globally. The rapid spread of antimicrobial-resistant pneumococcal isolates remains one of the global public health concerns [7]. Even the introduction of PVCs has been associated with an overall increase in the prevalence of antimicrobial-resistant serotypes not targeted by vaccination [5,7].

Since NP pneumococcal colonization is on the causal pathway to disease and a reflection of person-to-person transmission of the bacteria, we conducted this first Tunisian carriage study to investigate the prevalence of NP pneumococcal carriage and to assess the serotype distribution, antimicrobial susceptibility patterns, and evolutionary dynamics of NP isolates of *S. pneumoniae* in healthy children under five years in the south of Tunisia during a three-year survey period between 2020 and 2022.

## 2. Materials and Methods

The study’s approach was carried out following the World Health Organization Pneumococcal Carriage Working Group recommendations [8].

### 2.1. Study Population

A cross-sectional observational survey of asymptomatic NP *S. pneumoniae* carriage in healthy under-five children was performed in Sfax, southern Tunisia. This study was conducted over three survey periods, shortly after the introduction of PCV10 into the national immunization program: December 2020–June 2021, November 2021–June 2022, and October 2022–December 2022. A minimum of 100 carriage pneumococcal isolates per period was required. Each period ended when the target number of isolates was reached.

The study population included under-five children visiting vaccination centers for routine immunizations or attending kindergartens in the area of Sfax, the second biggest city in Tunisia (more than one million inhabitants). Exclusion criteria involved subjects suffering from ongoing acute upper or lower respiratory tract infection or who had received antibiotic treatment within the last 10 days before enrolment.

Informed consent was obtained from the parents of each child to participate in the study.

### 2.2. Sample Collection and Laboratory Methods

NP specimens were collected using flexible nylon-tipped flocked swabs (FLOQSwab^®^, COPAN). After collection, swab tips were placed in 1 mL skim milk tryptone–glucose–glycerol (STGG) transport medium. Samples were transported to the microbiology laboratory of Habib Bourguiba University Hospital, Sfax, Tunisia.

A questionnaire was used to collect information on demographic characteristics, PCV vaccination status, medical history, and factors that can be associated with pneumococcal NP carriage by interviewing parents and checking the vaccination cards. The vaccination status of the children was defined as unvaccinated, incomplete PCV vaccination (one or two doses of vaccine), and complete PCV vaccination (three or more doses of vaccine).

STGG specimens were inoculated on 5% defibrinated sheep blood agar, blood–colistin and nalidixic acid (CNA) agar, chocolate agar, and Todd Hewitt broth for enrichment culture. Plates were examined for the appearance of alpha-hemolytic colonies. For each sample, colonies with different morphologies were analyzed separately.

Isolates were identified as *S. pneumonia* based on standard laboratory procedures including Gram staining, colony morphology, optochin (5 µg) susceptibility, and bile solubility testing. Identification was confirmed by polymerase chain reaction (PCR) targeting the gene encoding autolysin (lytA).

Serotyping was conducted by latex agglutination (ImmuLex™ Pneumotest) and sequential multiplex PCR using primers, as described previously [9]. A primer pair amplifying the capsular polysaccharide biosynthesis gene (cpsA) was used as an internal control for the PCR reactions [9]. Each serotype determined by multiplex PCR was confirmed using a simplex PCR. Isolates that could not be serotyped by multiplex PCR reactions were serotyped by simplex PCR using primers targeting other serotypes, as published previously [10]. Isolates determined as 6A/B or 9V/A by the multiplex PCR method were typed to the serotype level using pneumococcal capsule-specific antisera (ImmuLex^TM^ Pneumotest). Confirmed pneumococcal isolates that were negative for the *cpsA* gene were considered non-encapsulated *S. pneumoniae* (NESp) according to a recent publication [11].

Isolates were tested for antimicrobial susceptibility by disk diffusion method including oxacillin (1 µg), erythromycin (15 µg), clindamycin (2 µg), pristinamycin (15 µg), gentamicin (500 µg), chloramphenicol (30 µg), tetracycline (30 µg), norfloxacin (10 µg), trimethoprim/sulfamethoxazole (1.25–23.75 µg), rifampicin (5 µg), vancomycin (5 µg), and teicoplanin (30 µg). The minimal inhibitory concentrations (MICs) of penicillin G, amoxicillin, cefotaxime, and levofloxacin were determined by E-tests. The two methods were performed on Mueller–Hinton agar supplemented with 5% sheep blood according to the Antibiogram Committee of the French Society for Microbiology and the European Committee on Antimicrobial Susceptibility Testing (CA-SFM/EUCAST). Categorically, based on the CA-SFM/EUCAST 2022 breakpoints, results were interpreted as ‘susceptible, standard dosing regimen’, ‘susceptible, increased exposure’, or ‘resistant’. The erythromycin–clindamycin double disk diffusion method was used to detect macrolide resistance phenotypes. Multidrug resistance (MDR) was defined as resistance to three or more classes of antimicrobials [12]. *S. pneumoniae* ATCC 49619 was used as a quality control strain.

### 2.3. Data Analysis

Statistical analyses were performed using IBM SPSS Statistics version 26.0. Pearson Chi-square and Fisher’s exact tests were used for the comparison of categorical variables. Changes over the three survey periods were assessed using the Chi-square test for trend.

We used univariate and multivariate logistic regression models to identify factors independently associated with pneumococcal NP carriage. Variables with *p*-value < 0.2 in the univariate analysis were included in the multivariate model. Odds ratios (ORs) with 95% confidence intervals (CI95%) were calculated to measure the association between potential associated factors and the occurrence of NP carriage.

Univariate analysis was carried out to determine the association between the serotype and antimicrobial resistance of *S. pneumoniae* isolates.

A *p*-value of <0.05 was considered statistically significant.

## 3. Results

### 3.1. Prevalence of Pneumococcal Nasopharyngeal Carriage

A total of 1196 healthy children aged under five years were enrolled. Among them, 58 participated in two and 1 child in three survey periods. NP swabs were collected from 459 participants in the first, 425 in the second, and 372 in the third study period. The age of children ranged from 2 to 59 months with a mean age of 23.9 (standard deviation: 20.6) months. At the time of enrolment, 375 children (29.9%) were unvaccinated, 431 (34.3%) were incompletely vaccinated (424 with PCV10 and 7 with PCV13), and 450 (35.8%) were fully vaccinated (271 with PCV10 and 179 with PCV13).

Of the 1256 nasopharyngeal swabs, 306 were positive for *S. pneumoniae*, corresponding to an overall prevalence of pneumococcal NP carriage of 24.4% (CI95%: [22–26.8%]). Carriage prevalence in each survey period was 22.9% (CI95%: [19.1–26.7%]), 24.5% (CI95%: [20.4–28.6%]), and 26.1% (CI95%: [21.6–30.6%]), respectively.

For children who participated in two or three survey periods, the dynamics of pneumococcal carriage were as follows: negative–negative for 27 children, positive–negative for 19, positive–positive for nine, negative–positive for three, and negative–negative-positive for the child who had participated three times.

In multivariate analysis (Table 1), factors including age (being more than two years), the presence of siblings, a history of respiratory tract infections, and previous hospitalization were independently associated with pneumococcal NP carriage.

### 3.2. Serotype Distribution of S. pneumoniae Carriage Isolates

Among the colonized children, a total of 309 pneumococcal isolates were collected and represented 25 different serotypes. Nine isolates were NESp. The most common serotypes were 14 (14.9%), 19F (12%), 6B (10.4%), and 23F (7.4%), which are covered by the PCV10 vaccine, as well as 19A (8.4%) and 6A (7.8%), which are covered by the PCV13 vaccine. Serotypes 1, 4, and 5 were not found. There was only one isolate of serotype 9V and two isolates of serotype 7F (Table 2). The theoretical vaccine coverages of PCV10 and PCV13 were 47.2% and 65.4%, respectively.

Figure 1 shows the distribution of serotypes according to the vaccination status of colonized children. The proportion of PCV10 serotypes in fully vaccinated children with PCV10 was significantly lower compared to unvaccinated children (37.7% vs. 53.7%, *p* = 0.043).

In this study, we identified three children co-colonized by two different isolates. The double serotype carriage was as follows: serotypes 19A and 23F, 3 and 14, and 6B and 6A. For children who tested positive in two survey periods, serotypes detected in the two periods were 19A/24F, 6C/35F, 19F/14, 13/14, 19F/35F, 14/6B, 10A/18C, 14/6B, and 6A/19A.

Between the three survey periods, a significant decrease in the proportion of serotype 19F (*p* = 0.001) and an increase in serotypes 19A (*p* = 0.034) and 6A (*p* = 0.029) were noted.

### 3.3. Antimicrobial Resistance of S. pneumoniae Carriage Isolates

Of the 309 *S. pneumoniae* isolates, resistance rates (R) to penicillin, amoxicillin intravenous (IV), and cefotaxime were detected in 4.9%, 14.9%, and 0.3%, respectively. Among tested antimicrobial agents other than beta-lactams, the highest rates of *S. pneumoniae* resistance were 78.3%, 68.9%, and 53.4% to erythromycin, clindamycin, and tetracycline, respectively. Regarding the resistance to macrolides, the following phenotypes were detected: constitutive macrolide–lincosamide–streptogramin B (cMLSB) phenotype for 211 isolates, inducible macrolide–lincosamide–streptogramin B (iMLSB) phenotype for 2 isolates, and the M phenotype for 29 isolates. Overall, multidrug resistance (MDR) was detected in 56.6% of the isolates. No acquired resistance was identified against levofloxacin, rifampicin, gentamicin, pristinamycin, vancomycin, and teicoplanin. Table 3 shows the antibiotic susceptibility of pneumococcal carriage isolates in the three survey periods.

The time trends of antimicrobial resistance rates throughout the three survey periods showed a significant increase in only trimethoprim/sulfamethoxazole (*p* = 0.008). No significant change in the rates of MDR *S. pneumoniae* was noted (*p* = 0.214).

Figure 2 shows the antibiotic resistance of the most frequent serotypes of *S. pneumoniae* carriage isolates. Serotype 14 was significantly associated with resistance to penicillin (OR: 114.6, CI95%: [14.5–900.8], *p*: <0.001), amoxicillin (OR: 9.1, CI95%: [4.4–18.6], *p*: <0.001), and trimethoprim/sulfamethoxazole (OR: 3.7, CI95%: [1.8–7.5], *p*: <0.001). All isolates of serotypes 19F and 19A were resistant to erythromycin. A significant association with resistance to this antibiotic was noted for serotypes 6B (OR: 9.7, CI95%: [1.3–72.4], *p*: 0.007) and 6A (OR: 6.9, CI95%: [1.0–52.3], *p*: 0.030). Serotypes 19F (OR: 19.1, CI95%: [4.5–81.1], *p*: <0.001), 6B (OR: 2.4, CI95%: [1.1–5.4], *p*: 0.027), 6A (OR: 2.8, CI95%: [1.1–7.3], *p*: 0.027), and 23A (OR: 8.6, CI95%: [1.9–38.1], *p*: 0.001) were significantly associated with resistance to tetracycline and 100% of 19A isolates were resistant to this antibiotic. MDR was detected in all 19A isolates and was associated with serotypes 19F (OR: 16.5, CI95%: [3.8–69.9], *p*: <0.001) and 23A (OR: 7.5, CI95%: [1.7–33.2], *p*: 0.002).

## 4. Discussion

This study provided new insights regarding the prevalence of pneumococcal carriage, serotype distribution, and antimicrobial susceptibilities of *S. pneumoniae* carriage isolates obtained from healthy children in the south of Tunisia during the early post-PCV10 period.

In the present study, almost one-quarter of under-five children carried *S. pneumoniae*. Many surveys assessing the prevalence of pneumococcal carriage have been conducted worldwide. NP pneumococcal carriage rates vary considerably between countries. Previous carriage studies in low- and lower-middle-income countries have described high NP *S. pneumoniae* colonization prevalence. According to a systematic review of the literature, the prevalence of pneumococcal carriage among children aged under five years in these countries ranged between 26.7% and 90.7% overall; 26.7 and 90.5% in pre-PCV studies; and 29.5 and 90.7% in post-PCV studies [13].

Factors affecting the variabilities in the burden of pneumococcal NP colonization include differences in the socio-demographic and clinical characteristics of the studied population, seasonality, geography, and detection methods [2,14]. The prevalence of carriage is strongly associated with accumulated risk factors. In our study, the main independent determinants associated with *S. pneumoniae* carriage were age (over two years), the presence of siblings, a history of respiratory tract infections, and previous hospitalization. Childcare attendance was associated with carriage only in univariate analysis. All of these determinants were previously described in the literature as risk factors for NP pneumococcal carriage [15]. The age group in which children have the highest colonization rate varies between studies. Overall, young age is an established risk factor for carriage [2,16].

It is known that pneumococcal vaccination leads to a decrease in VT carriage in vaccinated individuals and subsequently in unvaccinated individuals through herd immunity [2,15]. The potential role of vaccination in pneumococcal NP carriage is a critical issue [14]. Most studies worldwide have demonstrated that the overall pneumococcal carriage rate has remained similar in vaccinated and unvaccinated subjects. This is due to increased non-VT carriage rates among vaccinated children due to serotype replacement [14,17,18].

In the present study, no significant changes in the prevalence of NP pneumococcal carriage were observed in serial surveys conducted at short intervals following the implementation of PCV10. The most frequent serotypes (14, 19F, 6B, 19A, 6A, and 23F) were VTs. Carriage isolates belonged to PCV10 and PCV13 serotypes in 47.2% and 65.4%, respectively. These findings have previously been reported in the pre-vaccination period [19]. Some serotypes commonly account for the majority of nasopharyngeal carriage isolates from children. These include most of the serotypes represented in PCV7, as well as vaccine-related types 6A and 19A [19]. These serotypes were described in the literature as the most common serotypes of IPD cases, particularly in children under five years of age before PCV introduction, both in Africa [20] and globally [21]. Pneumococcal conjugate vaccines were designed to target these serotypes.

Cross-sectional studies may detect changes in the distribution of VT carriage as soon as a year post-PCV introduction if the sample size is sufficient [8]. Vaccine effects as profound changes in serotype distribution and herd protection may take three to four years to become apparent at high coverage [8,22]. Here, we have analyzed changes in serotype distribution during successive surveys conducted in the three years following the introduction of PCV10. A significant decrease in the proportion of serotype 19F and an increase in serotypes 19A and 6A were observed. This tendency has previously been discovered in children. Most studies worldwide have demonstrated that the carriage rate can remain the same by the shift toward non-VTs, particularly serotype 19A following PCV7 introduction [23,24]. The size of the changes varied between studies and can be related to the study population, vaccination coverage, and serotype distribution in the pre-PCV period [16].

In the current study, VT isolates were recovered even from children who had received the PCV10 or PCV13. The carriage of VTs by immunized children has been reported in the literature [24,25].

It has been shown through data from longitudinal studies that carriage is a dynamic process. Factors including age, previous pneumococcal exposure, and serotype determine the timing of acquisition and clearance [26]. The duration of carriage was reported to be longer for initial pneumococcal acquisitions than for subsequent ones, regardless of serotype [26]. This effect was explained not only by the maturation of the immune system with age but also by the fact that pneumococcal colonization is immunizing [18,26]. In our study, out of 59 children who participated in more than one survey period, 9 tested positive two times.

In addition to vaccine pressure, the serotype-specific prevalence in NP carriage has been explained, in substantial part, by the biochemical properties of the capsular polysaccharide. The structure of the capsule predicts the degree of encapsulation of different serotypes, their susceptibility to killing by neutrophils, and, ultimately, their success during nasopharyngeal carriage [27,28].

In the present study, three children were co-colonized by two different isolates. Simultaneous colonization by multiple pneumococcal serotypes is relatively common, especially in areas where the carriage rate and disease burden are high [8,18]. As standard methods underestimate multiple carriage, molecular serotyping directly from the specimen can ideally be used [8].

Among the 309 carriage isolates recovered from children in our study, nine were NESp. Recently, it has been reported that following PCV implementation, NESp, which lacks the cpsA gene, has emerged in many countries. A higher prevalence of NESp was observed in carriage or noninvasive isolates than in IPD clinical isolates. The spread and increased prevalence of NESp infections can be a concern in the post-PCV [11].

Since pneumococcal carriage precedes disease, antimicrobial resistance patterns of carried isolates may predict resistance rates among IPD and non-IPD. For beta-lactams, different susceptibility cut-offs (meningitis or other pneumococcal diseases) are recommended by the CA-SFM/EUCAST. Antimicrobial susceptibility testing of *S. pneumoniae* carriage isolates should use these different cut-offs as these isolates can serve as an indicator of the various pneumococcal diseases. When considering the cut-off value defining resistance for non-meningeal isolates, the resistance rates to penicillin G, amoxicillin, and cefotaxime were 4.9%, 14.9%, and 0.3% respectively. For macrolides, the resistance rate was alarming (78.3% for erythromycin). The high level of resistance to macrolides can be explained by the overuse of these antibiotics for children in our setting, as was noted when interviewing parents. A high rate of MDR was also observed during this study. A finding of concern in our study is the significant increase in trimethoprim/sulfamethoxazole resistance throughout the three survey periods.

In the current study, a significant association with antimicrobial resistance was observed for the most frequent serotypes of *S. pneumoniae* carriage isolates, mainly serotype 19A. According to the literature, the most common serotypes carried in children tend to be the most common resistant serotypes. It is in the nasopharynx where strains are likely to be exposed to prolonged antibiotic pressure and other commensal species with the ability of genetic transformation, leading to the acquisition of resistance [20,24,29].

Pneumococcal carriage in children is the principal source of resistance diffusion because children carry more often and for longer periods than adults and are frequently exposed to antibiotic use [24,30]. Before PCV introduction, most resistant serotypes were mainly VTs 6B, 9V, 14, 19F, 23F, and 6A [24]. Nevertheless, an increasing trend of resistant non-VT isolates has been observed in post-PCV surveillance studies, particularly serotype 19A following PCV7 introduction [5,24]. The association between VTs which were the most prevalent serotypes and antimicrobial nonsusceptibility has already been reported for isolates recovered from clinical samples in the pre-vaccination period in our setting [31].

In conclusion, this study is the first pneumococcal carriage survey in Tunisia. It has documented the relatively high prevalence of NP pneumococcal carriage in healthy children, the dominance of VTs, and the high level of macrolide resistance, penicillin-reduced susceptibility, and MDR rates among pneumococcal carriage isolates. An association between antimicrobial resistance and VTs has been demonstrated. An increase in non-PCV10 serotypes such as 19A and 6A following the implementation of PCV10 into our national immunization program has been noted. Thus, continuous monitoring of carriage is needed to evaluate changes in the prevalence of serotypes and antimicrobial resistance of pneumococcal isolates circulating in the post-PCV10 period. These changes must be taken into consideration when evaluating the Tunisian vaccination strategy.

## Figures and Tables

**Figure 1 vaccines-12-00393-f001:**
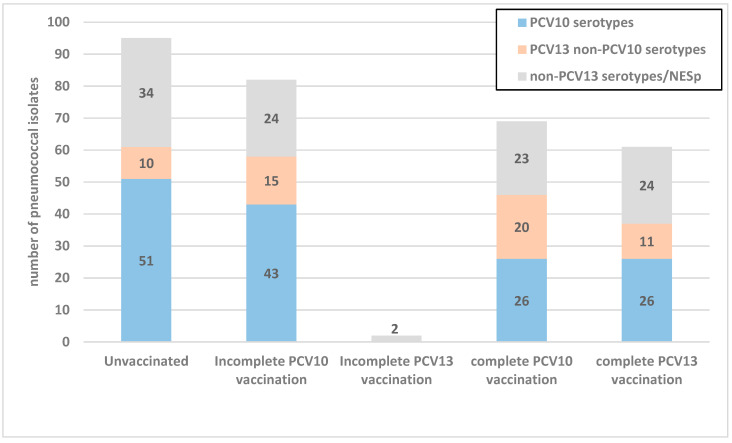
Vaccine serotype colonization according to pneumococcal conjugate vaccination status. Incomplete PCV vaccination—one or two doses of vaccine; complete PCV vaccination—three or more doses of vaccine; NESp—nonencapsulated *S. pneumoniae*.

**Figure 2 vaccines-12-00393-f002:**
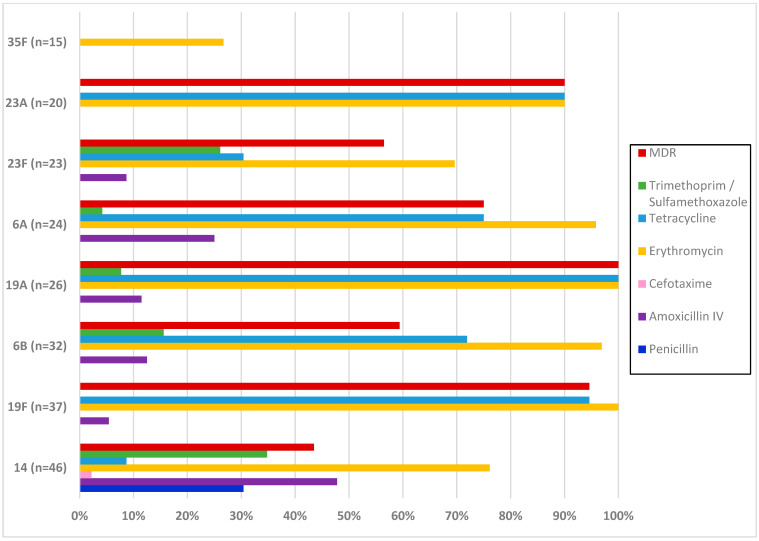
Antimicrobial resistance and multidrug resistance rates of the most frequent serotypes of *S. pneumoniae* carriage isolates. MDR—multidrug resistance; IV—intravenous; n—number of pneumococcal isolates.

**Table 1 vaccines-12-00393-t001:** Socio-demographic and medical factors associated with pneumococcal nasopharyngeal carriage.

Factors	Carriage Prevalence (%)	Univariate Model			Multivariate Model		
		OR	CI95%	*p*	OR	CI95%	*p*
Age, months							
2–24 (n = 779)	20.3	1.768	1.362–2.295	<0.001	1.570	1.052–2.345	0.027
25–59 (n = 477)	31						
Vaccination status							
Unvaccinated (n = 375)	25.3	1.00			1.00		
Incomplete PCV vaccination (n = 431)	19.3	0.703	0.503–0.982	0.039	0.821	0.524–1.287	0.390
Complete PCV vaccination (n = 450)	28.4	1.172	0.859–1.597	0.317	1.105	0.782–1.560	0.572
Presence of siblings							
Yes (n = 814)	27.4	1.632	1.228–2.168	0.001	1.485	1.099–2.006	0.010
No (n = 442)	18.8						
Childcare attendance							
Yes (n = 795)	28.2	1.813	1.364–2.410	<0.001	1.131	0.789–1.623	0.503
No (n = 461)	17.8						
History of respiratory tract infections							
Yes (n = 426)	32.4	1.888	1.450–2.459	<0.001	1.942	1.412–2.670	<0.001
No (n = 830)	20.2						
Antibiotic treatment within the last 3 months							
Yes (n = 409)	30.1	1.560	1.194–2.039	0.001	1.190	0.863–1.642	0.289
No (n = 847)	21.6						
History of hospitalization							
Yes (n = 102)	34.3	1.702	1.106–2.619	0.015	1.727	1.083–2.754	0.022
No (n = 1154)	23.5						

ORs—odds ratios; CI95%—95% confidence intervals; n—number of children; incomplete PCV vaccination—one or two doses of vaccine; complete PCV vaccination—three or more doses of vaccine.

**Table 2 vaccines-12-00393-t002:** Serotype distribution of *S. pneumoniae* carriage isolates in three survey periods.

Serotypes	Total n (%)	First Survey Period n (%)	Second Survey Period n (%)	Third Survey Period n (%)	*p*
PCV10 serotypes					
14	46 (14.9)	12 (11.4)	20 (19.2)	14 (14)	0.591
19F	37 (12)	22 (21)	9 (8.7)	6 (6)	0.001
6B	32 (10.4)	11 (10.5)	8 (7.7)	13 (13)	0.564
23F	23 (7.4)	5 (4.8)	8 (7.7)	10 (10)	0.153
18C	5 (1.6)	2 (1.9)	3 (2.9)	0 (0)	N/A
7F	2 (0.6)	0 (0)	0 (0)	2 (2)	N/A
9V	1 (0.3)	0 (0)	1 (1)	0 (0)	N/A
1	0 (0)	0 (0)	0 (0)	0 (0)	N/A
4	0 (0)	0 (0)	0 (0)	0 (0)	N/A
5	0 (0)	0 (0)	0 (0)	0 (0)	N/A
PCV13 non-PCV10 serotypes					
19A	26 (8.4)	5 (4.8)	8 (7.7)	13 (13)	0.034
6A	24 (7.8)	3 (2.9)	10 (9.6)	11 (11)	0.029
3	6 (1.9)	1 (1)	1 (1)	4 (4)	N/A
Non-PCV13 serotypes/NESp					
23A	20 (6.5)	5 (4.8)	14 (13.5)	1 (1)	0.297
35F	15 (4.9)	8 (7.6)	5 (4.8)	2 (2)	N/A
11A/D	13 (4.2)	3 (2.9)	3 (2.9)	7 (7)	N/A
24F	10 (3.2)	2 (1.9)	3 (2.9)	5 (5)	N/A
35B	9 (2.9)	5 (4.8)	1 (1)	3 (3)	N/A
16F	7 (2.3)	3 (2.9)	1 (1)	3 (3)	N/A
34	5 (1.6)	1 (1)	3 (2.9)	1 (1)	N/A
6C	5 (1.6)	2 (1.9)	2 (1.9)	1 (1)	N/A
10A	3 (1)	3 (2.9)	0 (0)	0 (0)	N/A
9N	3 (1)	0 (0)	2 (1.9)	1 (1)	N/A
13	2 (0.6)	1 (1)	0 (0)	1 (1)	N/A
15A	2 (0.6)	1 (1)	0 (0)	1 (1)	N/A
22F	2 (0.6)	1 (1)	1 (1)	0 (0)	N/A
15B/C	1 (0.3)	0 (0)	1 (1)	0 (0)	N/A
38	1 (0.3)	1 (1)	0 (0)	0 (0)	N/A
NESp	9 (2.9)	8 (7.6)	0 (0)	1 (1)	N/A
Total	309	105	104	100	-

NESp—nonencapsulated *S. pneumoniae*; N/A—not applicable.

**Table 3 vaccines-12-00393-t003:** Antimicrobial susceptibility of pneumococcal carriage isolates in the three survey periods (CA-SFM/EUCAST, 2022).

Antibiotic	Total			First Survey Period			Second Survey Period			Third Survey Period			*p*
	S (%)	I (%)	R (%)	S (%)	I (%)	R (%)	S (%)	I (%)	R (%)	S (%)	I (%)	R (%)	
Penicillin MIC: 0.06–2	15.9	79.3	4.9	16.2	81.9	1.9	12.5	79.8	7.7	19	76	5	N/A
Penicillin (meningitis) MIC: 0.06–0.06	15.9	-	84.1	16.2	-	83.8	12.5	-	87.5	19	-	81	
Amoxicillin IV MIC: 1–2	67.3	17.8	14.9	71.4	19	9.5	71.2	10.6	18.3	59	24	17	0.129
Amoxicillin IV (meningitis) MIC: 0.5–0.5	45.3	-	54.7	45.7	-	54.3	43.3	-	56.7	47	-	53	
Amoxicillin per os MIC: 0.5–1	45.3	22	32.7	45.7	25.7	28.6	43.3	27.9	28.8	47	12	41	
Cefotaxime MIC: 0.5–2	90.9	8.7	0.3	94.3	4.8	1	87.5	12.5	0	91	9	0	N/A
Cefotaxime (meningitis) MIC: 0.5–0.5	90.9	-	9.1	94.3	-	5.7	87.5	-	12.5	91	-	9	
Erythromycin	21.7	0	78.3	22.9	0	77.1	23.1	0	76.9	19	0	81	0.507
Clindamycin	31.1	-	68.9	36.2	-	63.8	30.8	-	69.2	26	-	74	0.115
Tetracycline	46.6	-	53.4	42.9	-	57.1	51	-	49	46	-	54	0.642
Trimethoprim/ Sulfamethoxazole	80.9	3.2	15.9	91.4	1	7.6	79.8	1	19.2	71	8	21	0.008
MDR	56.6			51.4			58.7			60			0.214

S—susceptible, standard dosing regimen; I—susceptible, increased exposure; R—resistant; MIC—minimal inhibitory concentration (µg/mL); IV—intravenous; MDR—multidrug resistance: resistance to three or more classes of antimicrobials; N/A—not applicable.

## Data Availability

The data presented in this study are available upon request from the corresponding author.

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
