# Peer review of "Nasopharyngeal Carriage of Streptococcus pneumoniae in Tunisian Healthy under-Five Children during a Three-Year Survey Period (2020 to 2022)"

_vaccines, 2024, doi:10.3390/vaccines12040393_

Round 1

Reviewer 1 Report

Comments and Suggestions for Authors

In the manuscript submitted by Ben Ayed, et al., the authors studied prevalence of nasopharyngeal pneumococcal carriage in a total of 1196 healthy children in Tunisia. Overall prevalence, serotype analysis, and antimicrobial resistance of the isolates were analyzed. The results are sound and the findings are interesting. The manuscript was generally well written.

Major Comments

The carriage of vaccinated serotypes for vaccinated children in this study is 37.7%, which seems higher than some other reports. This observation has been discussed (P10, L292-301) with a conclusion that PCV vaccination was not a significant protective factor against NP colonization by pneumococcus. This may not be consistent with the similar prevalence of most studies worldwide discussed in this paragraph, which was largely due to serotype replacement.

Minor Comments

1.     P1, Abstract. Add “In conclusion,” or “Conclusively, “ at the beginning of the last paragraph.

2.     P2, L55-56. This number should be updated to current, which is more than 100 serotypes.

3.     P4, L154-155. How? Please provide details.

4.     P7. L220-221. Clarify the indicated serotypes were detected in the two survey periods, respectively, but not co-colonized in both survey periods.

5.     P10, L299. “carriage prevalence” may be changed to “carriage rate”  to exclude prevalence of similar serotypes.

Author Response

Manuscript: vaccines-2900561

Dear Editor,

Thank you for considering our manuscript “Nasopharyngeal carriage of Streptococcus pneumoniae in Tunisian healthy under-five children during a three-year survey period (2020 to 2022).

Our detailed, point-by-point responses to the reviewer's comments are below. The changes in the manuscript are highlighted in yellow.

We hope that our revised manuscript will be accepted for publication.

We want to thank you once again for your consideration of our work and for inviting us to submit the revised manuscript.

Best regards,

POINT-BY-POINT RESPONSES TO COMMENTS FROM THE REVIEWER 1

Major Comments

The carriage of vaccinated serotypes for vaccinated children in this study is 37.7%, which seems higher than some other reports. This observation has been discussed (P10, L292-301) with a conclusion that PCV vaccination was not a significant protective factor against NP colonization by pneumococcus. This may not be consistent with the similar prevalence of most studies worldwide discussed in this paragraph, which was largely due to serotype replacement.

Response:

We have redrafted the paragraphs on page 10 from line 295 to line 339

Minor Comments

  1. P1, Abstract. Add “In conclusion,” or “Conclusively, “ at the beginning of the last paragraph.

Response: we have added ‶In conclusion″.

  1. P2, L55-56. This number should be updated to current, which is more than 100 serotypes.

Response: we have updated the number and added the recent reference (Manna, S.et al. Microbiol Spectr. 2024). The references list has been adjusted.

  1. P4, L154-155. How? Please provide details.

Response: we have modified the sentence ‶ Univariate analysis was carried out to determine the association between the serotype and antimicrobial resistance of S. pneumoniae isolates.″

  1. L220-221. Clarify the indicated serotypes were detected in the two survey periods, respectively, but not co-colonized in both survey periods.

Response: we have modified the sentence ‶ For children who tested positive in two survey periods, serotypes detected respectively in the two periods were 19A/24F, 6C/35F, 19F/14, 13/14, 19F/35F, 14/6B, 10A/18C, 14/6B, and 6A/19A. ″

  1. P10, L299. “carriage prevalence” may be changed to “carriage rate”  to exclude prevalence of similar serotypes.

Response: we have changed ‶carriage prevalence″ to ‶carriage rate″.

Reviewer 2 Report

Comments and Suggestions for Authors

- Authors should give more details about tree groups selection and criteria.

- During this period, it is very important to study the SARS-COV2 infection and coincidence, however this study didnot consider this, I can see the ethical approval was provided in 2018 but they should have modified the design, I wonder if it can be added. It is important and you can find hundreds of publication that can help.

- Vaccination status is not clear in methods (however in the manuscript authors includes children who visited vaccination center) and on results (table 1 as an example)

Comments on the Quality of English Language

Fine

Author Response

Manuscript: vaccines-2900561

Dear Editor,

Thank you for considering our manuscript “Nasopharyngeal carriage of Streptococcus pneumoniae in Tunisian healthy under-five children during a three-year survey period (2020 to 2022).

Our detailed, point-by-point responses to the reviewer's comments are below. The changes in the manuscript are highlighted in yellow.

We hope that our revised manuscript will be accepted for publication.

We want to thank you once again for your consideration of our work and for inviting us to submit the revised manuscript.

Best regards,

POINT-BY-POINT RESPONSES TO COMMENTS FROM THE REVIEWER 2

- Authors should give more details about tree groups selection and criteria.

Response: there were no specific criteria for the selection of the three groups. Our aim was to conduct three survey periods between 2020 and 2022 and each period ended when the target number of isolates (100 carriage pneumococcal isolates) was reached.

- During this period, it is very important to study the SARS-COV2 infection and coincidence, however this study didnot consider this, I can see the ethical approval was provided in 2018 but they should have modified the design, I wonder if it can be added. It is important and you can find hundreds of publication that can help.

Response: In this study, we have sought to assess the nasopharyngeal pneumococcal carriage in healthy children. Those suffering from ongoing acute upper or lower respiratory tract infections were excluded from the surveys.

- Vaccination status is not clear in methods (however in the manuscript authors includes children who visited vaccination center) and on results (table 1 as an example)

Response: The vaccination status of the children was determined as unvaccinated, incompletely vaccinated, and fully vaccinated according to the recommendations of the Advisory Committee on Immunization Practices (ACIP) https://www.cdc.gov/mmwr/preview/mmwrhtml/rr4909a1.htm.

We have added these definitions in the footnotes of table 1 ‶incomplete PCV vaccination, one or two doses of vaccine; complete PCV vaccination, three or more doses of vaccine.″

Reviewer 3 Report

Comments and Suggestions for Authors

This manuscript reports the prevalence of nasopharyngeal pneumococcal carriage and serotype distribution, along with its association with vaccinated conditions and antibiotic resistance of Streptococcus pneumoniae isolated from healthy under-five children in Tunisia. The authors’ work provides important information in the field of epidemiology, and the experimental procedures were well described and executed. I only have minor comments.

  1. 1. Section 3.2, Lines 197-232: The authors should provide insight into the selection of specific serotypes discussed in this section. Clarifying which serotypes correspond to PCV10 and/or PCV13 would enhance readability. Additionally, a brief description of NESp (non-encapsulated Streptococcus pneumoniae) would aid in comprehending the results.

  2.  
  3. 2. Incorporating Information in Tables and Figures: Including details about PCV10 and PCV13 serotypes in Table 2 and Figure 2 would facilitate interpretation for readers.

  4.  
  5. 3. Serotypes 1, 4, and 5 should be added to Table 2, despite their absence in the study, as they are relevant PCV serotypes. This addition ensures completeness and informs readers about all pertinent serotypes covered by PCV vaccines.

Author Response

Manuscript: vaccines-2900561

Dear Editor,

Thank you for considering our manuscript “Nasopharyngeal carriage of Streptococcus pneumoniae in Tunisian healthy under-five children during a three-year survey period (2020 to 2022).

Our detailed, point-by-point responses to the reviewer's comments are below. The changes in the manuscript are highlighted in yellow.

We hope that our revised manuscript will be accepted for publication.

We want to thank you once again for your consideration of our work and for inviting us to submit the revised manuscript.

Best regards,

POINT-BY-POINT RESPONSES TO COMMENTS FROM THE REVIEWER 3

  1. Section 3.2, Lines 197-232: The authors should provide insight into the selection of specific serotypes discussed in this section. Clarifying which serotypes correspond to PCV10 and/or PCV13 would enhance readability. Additionally, a brief description of NESp (non-encapsulated Streptococcus pneumoniae) would aid in comprehending the results.

Response: we have modified the sentence ‶ The most common serotypes were 14 (14.9%), 19F (12%), 6B (10.4%), and 23F (7.4%) which belonged to the PCV10 vaccine, as well as 19A (8.4%) and 6A (7.8%) which are covered by the PCV13 vaccine. ″ Lines 198-200.

We have already described the NESp in the Materials and Methods section, lines 129-130.

  1. Incorporating Information in Tables and Figures: Including details about PCV10 and PCV13 serotypes in Table 2 and Figure 2 would facilitate interpretation for readers.

Response: we have modified Table 2.

  1. Serotypes 1, 4, and 5 should be added to Table 2, despite their absence in the study, as they are relevant PCV serotypes. This addition ensures completeness and informs readers about all pertinent serotypes covered by PCV vaccines.

Response: we have added these serotypes to Table 2.